# Dynamic Performance of Suspended Pipelines with Permeable Wrappers under Solitary Waves

Youkou Dong [1], Enjin Zhao [1,*], Lan Cui [2], Yizhe Li [1] and Yang Wang [3]

1   College of Marine Science and Technology, China University of Geosciences, 388 Lumo Road, Wuhan 430074, China; dongyk@cug.edu.cn (Y.D.); yizheli@cug.edu.cn (Y.L.)
2   Institute of Rock and Soil Mechanics, Chinese Academy of Sciences, Wuhan 430071, China; lcui@whrsm.ac.cn
3   Haikou Marine Geological Survey Center, China Geological Survey, Haikou 570100, China; wangyang01@mail.cgs.gov.cn
*   Correspondence: zhaoej@cug.edu.cn

**Abstract:** Submarine pipelines are widely adopted around the world for transporting oil and gas from offshore fields. They tend to be severely ruined by the extreme waves induced by the natural disaster, such as hurricanes and tsunamis. To maintain the safety and function integrity of the pipelines, porous media have been used to wrap them from the external loads by the submarine environment. The functions of the porous wrappers under the hydrodynamic impact remain to be uncovered before they are widely accepted by the industry. In this study, a numerical wave tank is established with the immersed boundary method as one of the computational fluid dynamics. The submarine pipelines and their porous wrappers are two-way-coupled in terms of displacement and pressure at their interfaces. The impact from the solitary waves, which approximately represent the extreme waves in the reality, on the pipelines with different configurations of the porous wrapper is investigated. The results present significant protective functions of the wrappers on the internal pipelines, transferring the impact forces from the pipelines to the wrappers. The protective effects tend to be enhanced by the porosity and thickness of the wrappers. The influence of the pipeline configurations and the marine environment are then analysed. As for the front pipeline, an increase in the gap leads to a slight increase in the horizontal forces on both the wrapper and the pipeline, but a significant increase in the vertical forces. As for the rear pipeline, because of the shield function of the front pipeline, the velocity within the gap space and the forces on the pipes are decreased with the decrease in the gap size. The complex flow fields around the pipelines with wrappers are also illuminated, implying that the protection function of the wrapper is enhanced by the wave height reduction.

**Keywords:** extreme wave; submarine pipeline; external wrapper; coupling analysis; computational fluid dynamics



## 1. Introduction

Pipelines that are laid on or below the seabed and continuously transport large amounts of oil (or gas) are collectively referred to as submarine pipelines. They constitute the main transporting structures and currently they are the most economical and reliable selections in the design of transportation tools. Pipelines are usually installed within the seabed sediments under the protection of rock berms [1]. However, the sediments around the pipelines may be scoured by contour currents and internal waves, which expose the pipelines to the threat of complex marine environments [2]. The scour mechanism and its evolution process around the in-position pipelines were investigated by many scholars, such as Reference [3]. Occasionally, segments of a pipeline may be suspended between high points through continental slopes due to an uneven seabed profile. For example, suspended pipelines were widely used in the Ormen Lange projects, with massive depressions and landslide blocks scattered along the 120-km-long route [4].

Natural disaster, such as hurricanes and tsunamis, may induce extreme waves that generate enormous impact loads on the pipelines and may cause serious ruins to the whole production and transportation system [5–7]. Tsunamis, one of the major marine disasters caused by earthquakes and submarine landslides [8,9], send surges of water with extremely long waves that are not especially steep [10]. The tsunami triggered by a 9.0-Mw earthquake in 2011 extensively destroyed 70% of the total 200,000 structures along the Miyagi coastline, including submarine pipelines, seawalls, and coastal bridges. A tsunami is typically composed of several transient waves with varying amplitudes, wavelengths, and wave periods during propagation. Solitary waves were proposed to simulate the tsunami waves by decomposing them into N-waves through the Korteweg-de Vries equation [11–14]. Since then, the run-up process of the tsunami waves along the shoreline was investigated with the depth-averaged smooth particle hydrodynamics method [15,16]. References [17,18] quantified the impact loads over cylinders from a tsunami wave.

To protect the marine structures from potential damages due to extreme marine conditions, engineers have developed outer protections in terms of wrappers made of porous media. A porous medium enhances the buffering performance of the structures and dissipates part of the incoming wave energy [19]. For example, the turbulent intensities on a permeable breakwater were significantly attenuated in the numerical analysis by References [20–22]. Naturally, porous media are expected to be protective to submarine pipelines under extreme marine conditions, although thermal insulation and erosion prevention were mainly considered in designing pipeline coatings in the industry [23,24]. Reference [25] quantified the wave forces on pipelines buried in an impermeable bed with coverings of porous media. References [26,27] evaluated the protective performance of a porous polymer coating on subsea pipelines under sudden impacts. The drag reduction function of the porous coatings over cylinders were then quantified by Reference [28]. Two factors were considered to influence the stabilization effect of the porous coatings on pipelines: the production of an entrainment layer through the coating and the triggering of turbulent transition of the detaching shear layers. In engineering practice, applications of porous coatings on submarine pipelines are limited. Concrete wrappers, mainly designed to counteract the buoyancy forces of pipelines, can be considered as one kind of porous wrapper with medium permeability. In addition, porous wrappers made with woven carbon-fiber materials or polyurethane foam may be designed in future for pipeline protection.

The above literature review revealed that few studies were performed to examine the protective effect by the porous media on submarine pipelines, which is the main aim of this study. The porous wrapper and the submarine pipeline modules are simulated in a numerical wave tank (NWT) with the immersed boundary (IB) method. The numerical methods and equations will be provided in Section 2. Verification of the numerical model is provided in Section 3. The parametric simulations are in Section 4, in which the effects of different waves on various pipelines with porous wrappers are analysed. The conclusions are given in Section 5.

## 2. Numerical Methods

For simulating the interactions between pipelines and waves, the finite volume methods have been widely used. In this study, the commercial finite volume package FLOW-3D® (version 11.1.0; 2014; https://www.flow3d.com (accessed on 10 December 2022); Flow Science, Inc., Santa Fe, NM, USA). Flow-3D aims to solve the transient response of fluids under interactions with structures, internal and external loads and multi-physical processes. It features some advantages in terms of a high level of accuracy in solving the Navier-Stokes equation with the volume of fluid (VOF) method, efficient meshing techniques for complex geometries, and high efficiency level for large-scale problems. Also, Flow-3D provides the flexibility and utility for flowing through porous media. A two-dimensional numerical wave tank was constructed by using the immersed boundary (IB) method and an in-house subroutine termed as IFS_IB. A submarine pipeline and porous medium were two-way coupled at the interface described by the individual volume fractions [29]. The pipeline

was wrapped with a layer of a porous medium. A solitary wave was generated at the inlet boundary of the tank to simulate an approaching tsunami. Non-slip wall conditions were assigned at the bottom of the tank and the pipe surface, which was also specified with a roughness coefficient. The top boundary was defined as a free boundary and configured with the atmospheric pressure. A Neumann-type absorbing boundary condition, a stable, local, and absorbing numerical boundary condition for discretized transport equations [30], was imposed on the outlet boundary to attenuate the reflections of the outgoing waves. A transition zone is set within a certain range from the boundary to reduce the horizontal gradient force of the elements near the boundary and suppress the calculation wave caused by this boundary condition. Through the relaxation coefficient, the predicted value on the inner boundary of the transition zone and the initial value on the outer boundary are continuously transitioned to achieve the purpose of reducing the reflection of propagating waves. The CUSTOMIZATION function of the software FLOW-3D was utilised to impose the Neumann-type absorbing boundary condition. The FLOW-3D distribution includes a variety of FORTRAN source subroutines that allow the user to customize FLOW-3D to meet their requirements. The FORTRAN subroutines provided allow the user to customize boundary conditions, include their own material property correlations, specify custom fluid forces (i.e., electromagnetic forces), add physical models to FLOW-3D, and have additional benefits. Several "dummy" variables have been provided in the input file namelists that users may use for custom options. A user definable namelist has also been provided for customization. Makefiles are provided for Linux and Windows distributions and Visual Studio solution files are provided for Windows distributions to allow users to recompile the FLOW-3D code with their customizations.

### 2.1. Governing Equations

The governing equations involved include the continuity equations and Reynolds-averaged Navier-Stokes equations. The mass and momentum are conserved in a two-dimensional zone [31]:

$$\frac{\partial \rho}{\partial t} + \nabla \cdot (\rho \mathbf{U}) = 0 \tag{1}$$

$$\frac{\partial (\rho \mathbf{U})}{\partial t} + \nabla \cdot (\rho \mathbf{U}\mathbf{U}) = -\nabla P + \mathbf{g} \cdot \mathbf{X} \nabla \rho + \mu \nabla^2 \cdot \mathbf{U} + \sigma \kappa \nabla \alpha \tag{2}$$

where $\mathbf{U}$ is the velocity vector, $\mathbf{X}$ is the Cartesian position vector, $\mathbf{g}$ denotes the gravitational acceleration vector, and $\rho$ represents the weighted averaged density. The term $\mu$ is the viscosity. $\sigma \kappa \nabla \alpha$ identifies the surface tension effects with $\sigma$ as the surface tension and $\alpha$ as the fluid volume fraction. Each cell in the fluid domain has a water volume fraction ($\alpha$) ranging between 0 and 1, where 1 represents cells that are fully occupied with water, while 0 represents cells that fully occupied with air. Values between 1 and 0 represent free surface between air and water. The free surface elevation is defined by using the volume of fluid (VOF) function:

$$\frac{\partial F}{\partial t} + \frac{1}{V_F} \left[ \frac{\partial}{\partial x} (\alpha A_x u) + R \frac{\partial}{\partial y} (\alpha A_y v) + \frac{\partial}{\partial z} (\alpha A_z w) + \varepsilon \frac{F A_x u}{x} \right] = F_{DIF} + F_{SOR} \tag{3}$$

where $V_F$ is the volume of fluid fraction, $F_{SOR}$ is the source function, $F_{DIF}$ is the diffusion function; $A_x$, $A_y$, and $A_z$ represent the fractional areas; and $u$, $v$, and $w$ are the velocity components in the $x$, $y$, and $z$ directions.

### 2.2. Porous Media Module

In FLOW-3D, the porous medium's flow resistance is modelled by the inclusion of a drag term in the momentum equations (Equation (2)). Coarse granular material is used in most coastal engineering applications, in which case the Forchheimer model is suitable. Using this model, a drag term $F_d u_i$ is added to the righthand-side of Equation (2):

$$F_d\mathbf{U} = -g\left(an\mathbf{U} + bn^2|\mathbf{U}|\mathbf{U}\right) \tag{4}$$

where $|\mathbf{U}|$ is the norm of the velocity vector, $n$ the porosity, and $a$ and $b$ are the factors.

*2.3. Solitary Wave Boundary*

The solitary wave is generated in terms of variations of the surface elevation $\eta$ and velocities $u$ and $v$ by following McCowan's theory [32]:

$$\eta = Qh \tag{5}$$

$$u = c_0 E \frac{1 + \cos\left(M\frac{z}{h}\right)\cosh\left(M\frac{x}{h}\right)}{\left[\cos\left(M\frac{z}{h}\right) + \cosh\left(M\frac{x}{h}\right)\right]^2} \quad v = c_0 E \frac{\sin\left(M\frac{z}{h}\right)\sinh\left(M\frac{x}{h}\right)}{\left[\cos\left(M\frac{z}{h}\right) + \cosh\left(M\frac{x}{h}\right)\right]^2} \tag{6}$$

where $h$ is the still water depth; $Q$ is the reference value

$$Q = \frac{E}{M} \frac{\sin[M(1+Q)]}{\cos[M(1+Q)] + \cosh(+X/h)} \tag{7}$$

$$E = \frac{2}{3}\sin^2[M(1 + \frac{2H}{3h})] \quad M = E\frac{h}{H}\tan[\frac{1}{2}M(1 + \frac{H}{h})] \tag{8}$$

where $X = x - c_0 t$; $c_0 = g\sqrt{H+h}$; $H$ is the wave height; and $t$ is the elapsed time.

## 3. Validation

### 3.1. Propagation over a Porous Breakwater

An experimental test on the propagation process of a solitary wave over a permeable breakwater was performed by Reference [20], which was simulated in this study to validate the adopted two-way coupling model (Figure 1a). The length, width, and depth of the flume tank were 25, 0.5, and 0.6 m, respectively. A permeable breakwater was mounted at the bottom of the flume, which had dimensions of 13 cm and 6.5 cm in the length and height, respectively. The porous breakwater with an average porosity of 0.52 was configured by glass beads with a constant diameter of 1.5 cm. Two wave gauges were fixed before (WG$_1$) and behind (WG$_2$) the breakwater, respectively. The initial still water depth $h$ was assumed to be 10.6 cm. Height of the solitary wave $H$ was considered to be 4.77 cm. In the numerical model, the calculation zone had dimensions of 5 m in length and 0.25 m in height. The second order quadrilateral mesh elements were adopted. The grid around the breakwater was the finest of 0.001 m. The adopted time step size was 0.05 s. The numerical predictions of the water elevations at the locations WG$_1$ and WG$_2$ by the adopted numerical tool FLOW-3D are close to both the experimental measurements and the numerical predictions from another CFD FLUENT version 14.0.1 [33] (Figure 1). Figure 1b,c show the comparison of monitored water levels at the two water level monitoring points in Figure 1a. It can be seen that the experimental results of the two monitoring points are consistent with the numerical simulation results, indicating that the propagating solitary wave energy is basically completely dissipated and then flows out. If the propagating wave energy is not dissipated, the phenomenon of wave reflection will occur. The waves monitored at the two monitoring points will appear superposition of propagating waves and reflected waves. The numerical simulation results do not agree with the physical experiment results. The fluctuations of the water surface elevation after the bypass of the incoming wave are due to its residual reflection at the right absorbing boundary condition, which arrives at WG$_2$ at an earlier time than WG$_1$. Evolution of the wave surfaces was also compared between the experimental and the numerical models (Figure 2), which demonstrates that the numerical tool is sufficiently reliable. The velocity of the wave is reduced by the porous medium as it partially infiltrates into the breakwater, which is shown as in Figure 3 by comparing the horizontal velocity distributions between the experimental and numerical results at times

of 1.5 s and 2 s. The numerical predictions of the flow velocities have slight discrepancies with the experimental measurements, which are attributed to the material assumptions made in the numerical model for the glass beads in the experimental setup.

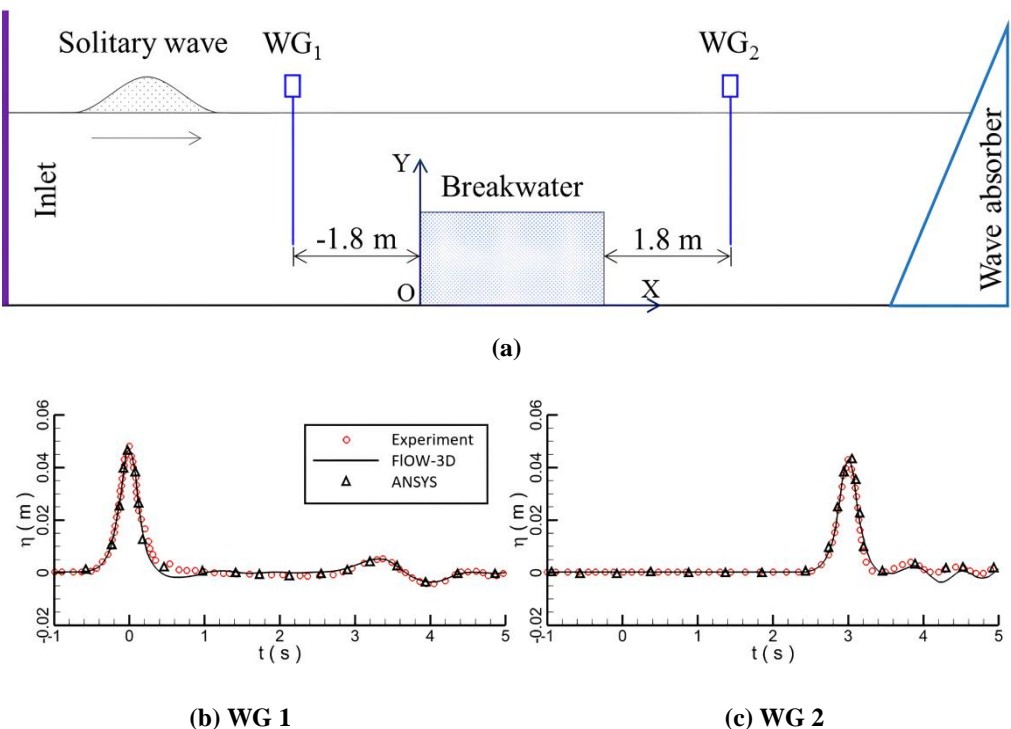

**Figure 1.** The diagrammatic sketch of the numerical setup (non–scaled) (**a**) and the temporal evolution comparison of water surface between experimental and numerical results (**b**,**c**).

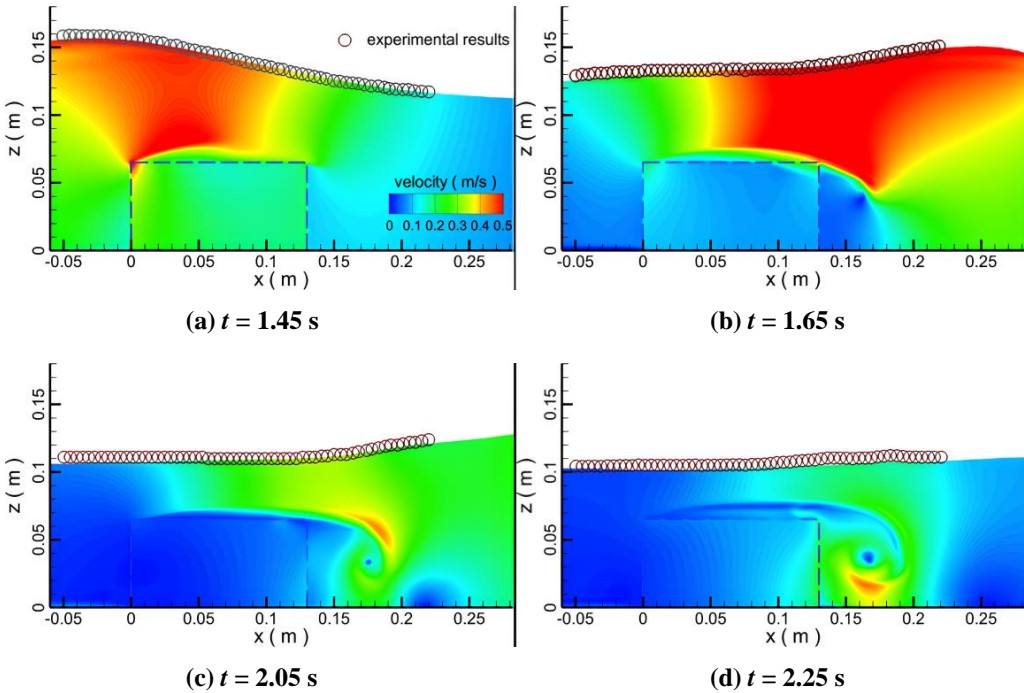

**Figure 2.** Water surface comparison between experimental and numerical results.

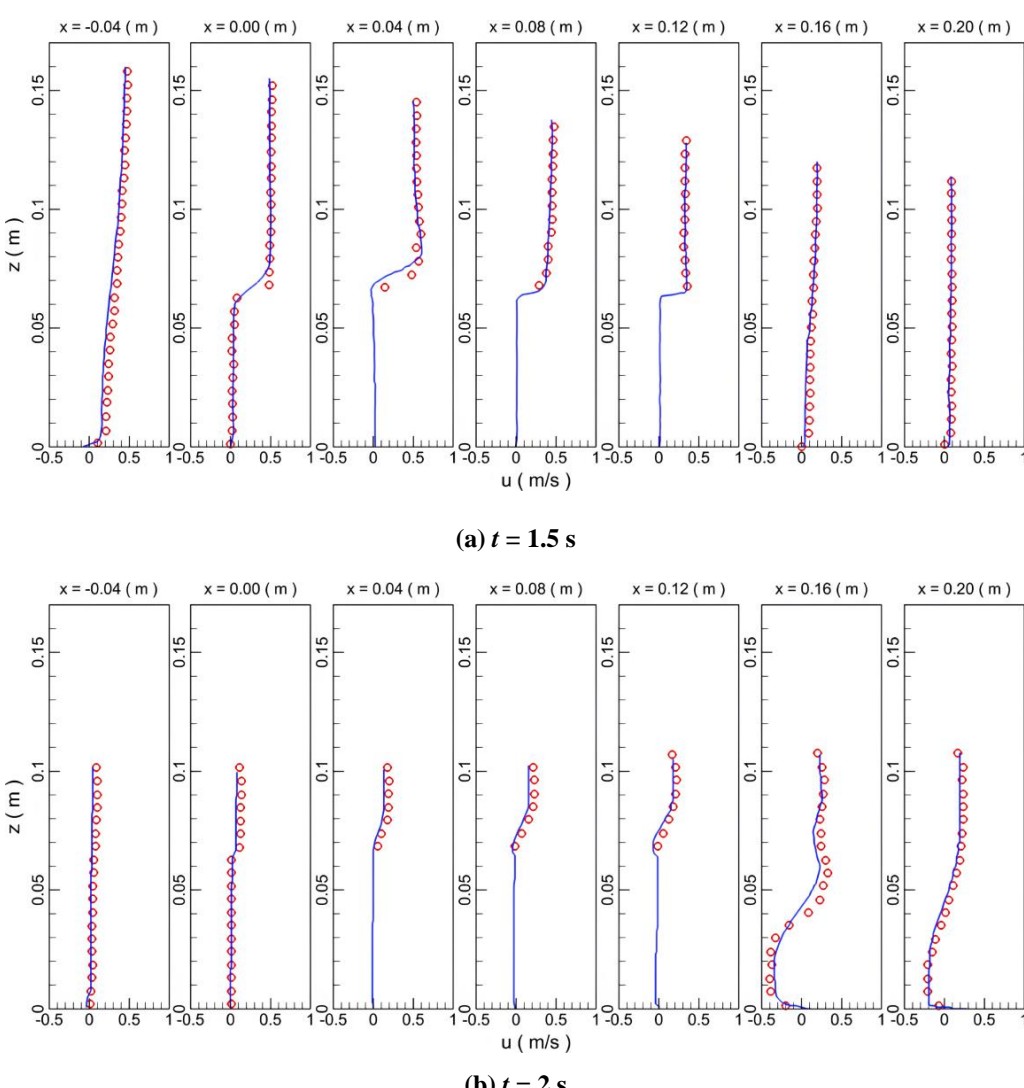

**(a)** *t* = 1.5 s

**(b)** *t* = 2 s

**Figure 3.** Comparison of horizontal velocity distribution between experimental and numerical results.

### 3.2. Forces on Pipeline

Another experimental test of a solitary wave impacting a pipeline was performed by Reference [34], which was also reproduced in this study for validation purposes. The calculation zone had dimensions of 40 m in length and 0.6 m in height. The solitary wave had a height of 0.0555 m with the initial water depth of 0.192 m. The pipe had a diameter of 0.048 m, which had a distance of 0.136 m over the bottom boundary of the model. A dense mesh consisting of 413,411 cells was employed with a mesh size of 0.1 mm around the pipe, which proved to be sufficiently fine through convergence studies. History of the horizontal and vertical forces, normalized by $\rho g L(\pi D^2/4)$ with $L$ as the unit length of 1 m, is compared between the experimental and numerical results (Figure 4). Both the peak values and the transient variations of the forces predicted by the numerical analysis converge to the measured values in the experimental test. The slight discrepancy between the numerical and experimental results at 2.5 s and 3.1 s, which may be induced by the error of the numerical model simulating the complicated turbulence behaviour, is acceptable in relation to the requirements of this study as our concern is mainly the peak values of forces.

Therefore, the adopted numerical tool is sufficiently reliable to simulate the interactions between solitary waves and the permeable structure through the above validation cases.

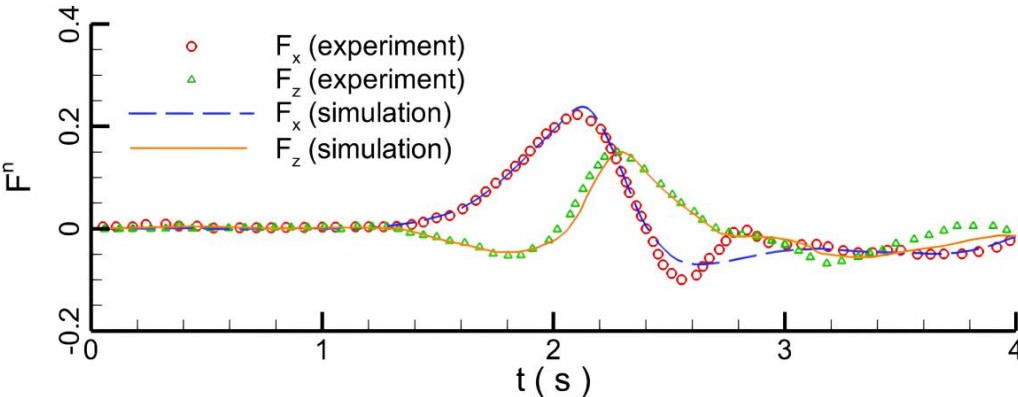

**Figure 4.** Force comparison between the experimental and numerical results.

## 4. Results and Discussion

Influence of the solitary waves on the performance of wrapped pipelines was investigated by considering different wave heights (*H*) and thicknesses (*T*) and wrapper porosities (*n*). The still water depth (*h*) was taken to be 6 m (Figure 5). The diameter of the porous medium was assumed to be 0.05 m. The pipeline diameter *D* was set at 1 m. In Figure 5 the variable *G* represents the gap between the permeable wrapper and the seabed. The scouring process had been completed before the simulation; therefore, the seabed boundary was taken as a rigid wall. The tandem pipelines had a distance of *S* between each other. The whole model had dimensions of 400 m in length and 12 m in height. The finest mesh around the pipeline was configured as 0.0025 m, which was verified to be sufficiently fine through trial calculations with finer meshes.

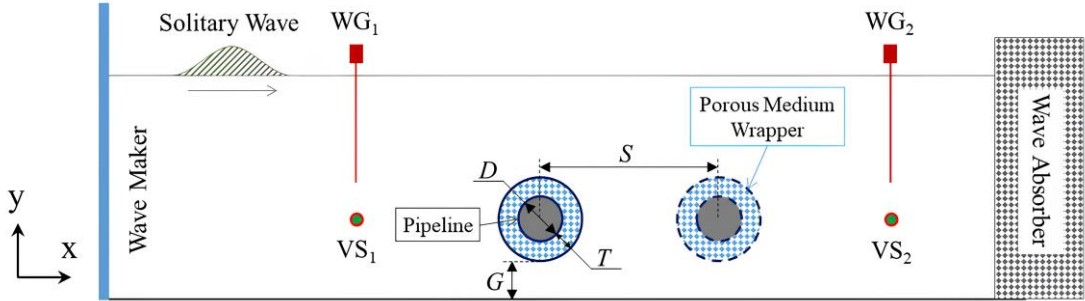

**Figure 5.** Layout for solitary wave impinging on the submarine pipeline encased in porous media.

### 4.1. Effect of Porous Wrapper

#### 4.1.1. Wrapper Porosity

The pipeline was put on the seabed. The gap (*G*) between the wrapper and the seabed was considered to be zero. The height (*H*) of the solitary wave was considered to be 2 m. The porosity (*n*) was taken to be 0.0, 0.4, 0.6, and 1.0. Note that *n* = 0.0 indicates the impervious condition, while *n* = 1.0 corresponds to the non-wrapping condition. The thickness of the permeable wrapper remained at 0.5 m. In calculation, the wave approaches the pipe at around 6.3 s and departs from it at 10.2 s. When the wave approaches, the kinematic performance over the pipe is enhanced (Figure 6). Due to the wave disturbance, a number of small vortices are generated around the pipe (Figure 7). At the departure of the wave, the disturbance to the flow field seems to be more intense than that at its arrival, which further generates vortices around the pipeline. Without a wrapper, the pipe is fully exposed to the disturbance of the incoming wave, which maximises the velocity and vorticity values around the pipe. When the pipeline is wrapped by a porous medium, some water seeps into the wrapper, and the velocity in the wrapper is reduced to a very low

value, which implies that the porous medium is capable of absorbing the dynamic energy of the flowing fluid. With an external coverage ($n < 1.0$), the disturbance is generated mainly at the outer surface of the wrapper. As the wrapper porosity increases, the domain of the low-speed flow underneath the pipeline expands.

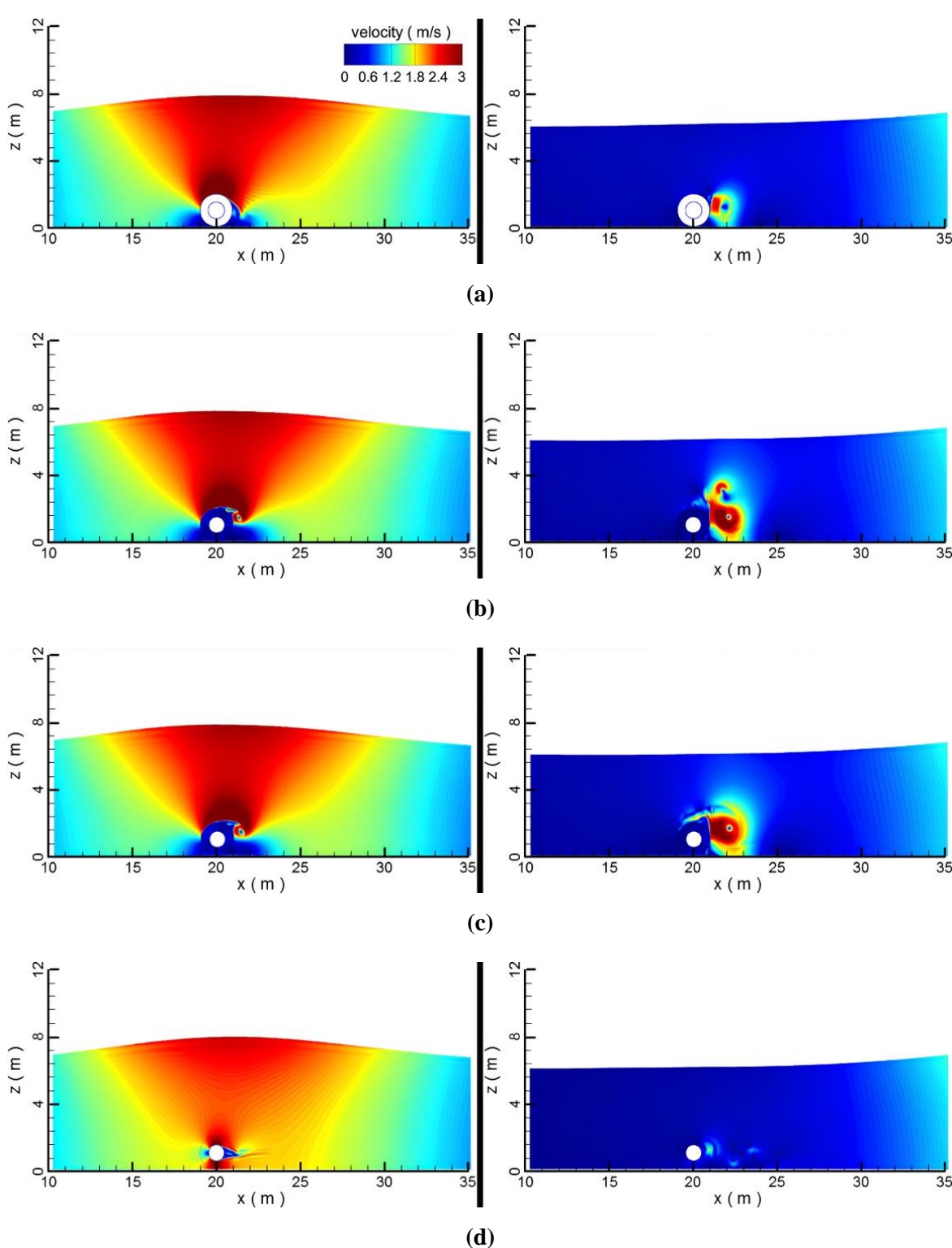

**Figure 6.** The velocity contours of the flow fields under different porosities; (**a**) $n = 0.0$; (**b**) $n = 0.4$; (**c**) $n = 0.6$; (**d**) $n = 1.0$; left to right: arrival, departure.

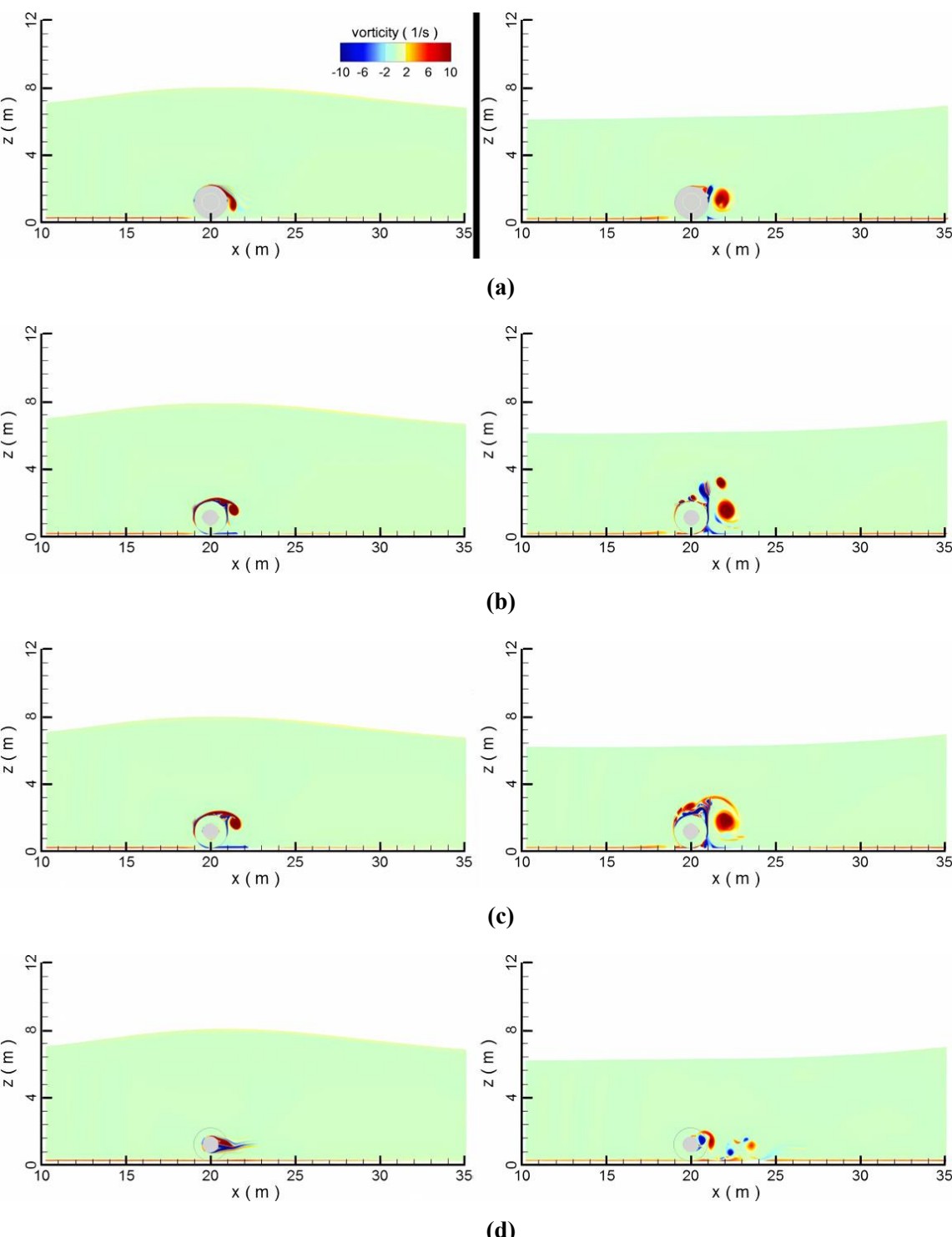

**Figure 7.** The vorticity contours of the flow fields under different porosities; (**a**) $n = 0.0$; (**b**) $n = 0.4$; (**c**) $n = 0.6$; (**d**) $n = 1.0$; left to right: arrival, departure.

The peak velocity around the pipeline without a wrapper (1.9 m/s) is larger than that with a wrapper (1.6 m/s) (Figure 8). For pipes with wrappers, the peak velocities around them are similar to one another. In contrast, the velocity profiles at x = 23 m are quite different. When the pipeline has no wrapper (i.e., $n = 1.0$), the change in velocity is fairly moderate. When the pipeline has a wrapper, the porous wrapper causes a secondary fluctuation in the rear water body after the primary fluctuation due to the peak of the wave

passing through the pipeline. This generates a series of velocity peaks. The secondary velocity peaks for a porosity coefficient of 0.4 are higher than those for a porosity coefficient of 0.6. Accordingly, the turbulent kinetic energy (TKE) also changes with the porosities, as shown in Figure 9. The TKE is expressed as

$$\text{TKE} = \int \frac{1}{2}\rho(|u|^2 + |v|^2)dV_f \tag{9}$$

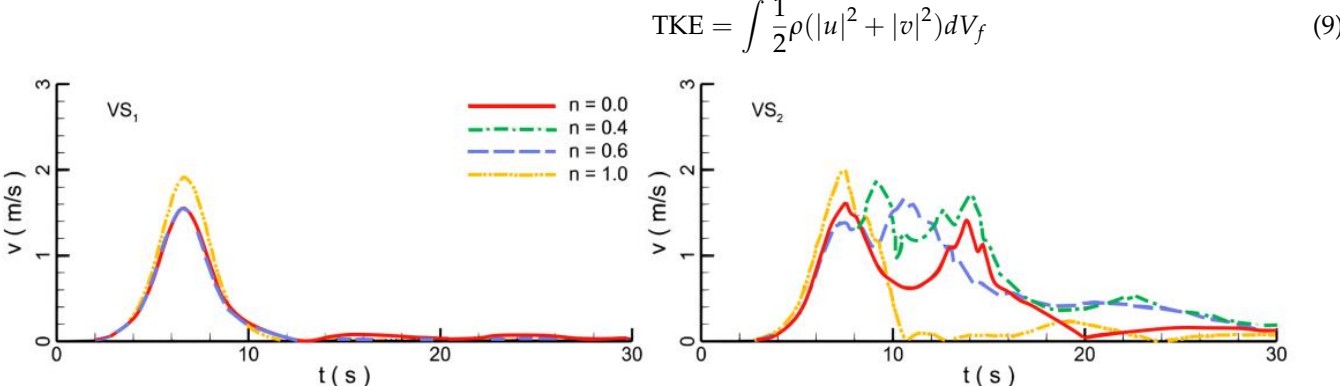

**Figure 8.** Comparison of horizontal and vertical velocities at front and rear of pipeline under different porosities.

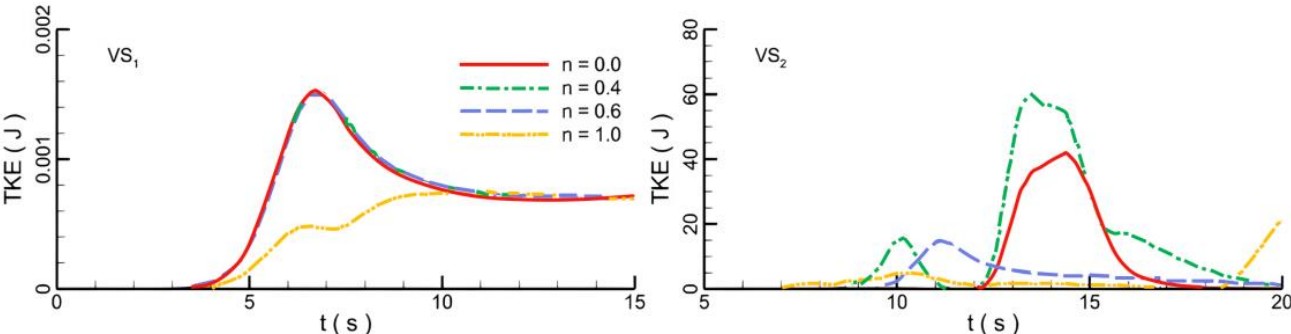

**Figure 9.** Comparison of turbulent kinetic energy at front and rear of wrapper under different porosities.

With the propagation of the wave, the TKE increases gradually in front of the pipeline. The TKE value under the pipeline without a wrapper ($n = 1.0$) (0.0008 kJ) is nearly half of that with a wrapper (0.0015 kJ). In comparison, the TKE values for the wrapped pipelines ($n < 1.0$) are very close to each other. After the wave leaves the pipeline, the TKE in front of the pipeline decreases for around 50%. Then, the TKE in the rear of the pipeline with a porous wrapper increases intensively because the porous media perturb the flow field. Compared with the pipeline without the wrapper, the interaction between the wrapped pipeline with the flow field is more severe. Furthermore, the solid wrapper can cause a strong disturbance to the flow, but the interference of the solid wrapper ($n = 0.0$) in the rear flow is still weaker than the wrapper with the porosity of 0.4.

The hydrodynamic forces ($F$), including the pressure and shear stress, are normalized by $\rho g L(\pi D^2/4)$ (Figure 10). With a fully solid (i.e., $n = 0.0$) wrapper, the pipeline tends to be unaffected by the external flow. Hence, the hydrodynamic forces are zero while the forces on the wrapper reach their maximum. With porous wrappers, water seeps into the wrapper, buffering the impact of the incoming waves on the pipe. As the porosity coefficient increases, the induced forces on the pipeline increase while those on the wrapper decrease. When the porosity coefficient is 0.4, the forces on the external wrapper become higher than that on the internal pipeline. Therefore, the porous wrapper is capable of protecting the pipeline. The smaller the porosity coefficient the better protection the wrapper provides to the pipeline. The pressure gradient and shear stress forces are also shown in Figure 11.

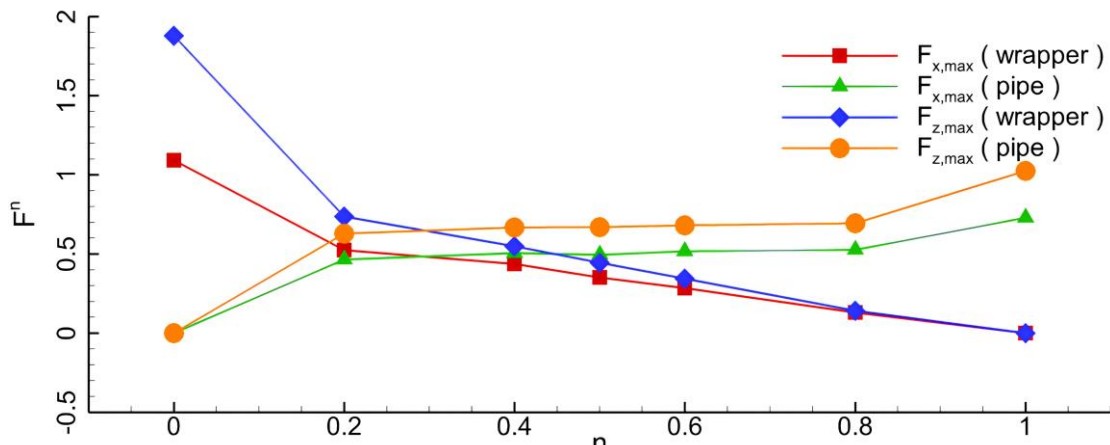

**Figure 10.** Comparisons of the maximum hydrodynamic forces on the pipeline and wrapper.

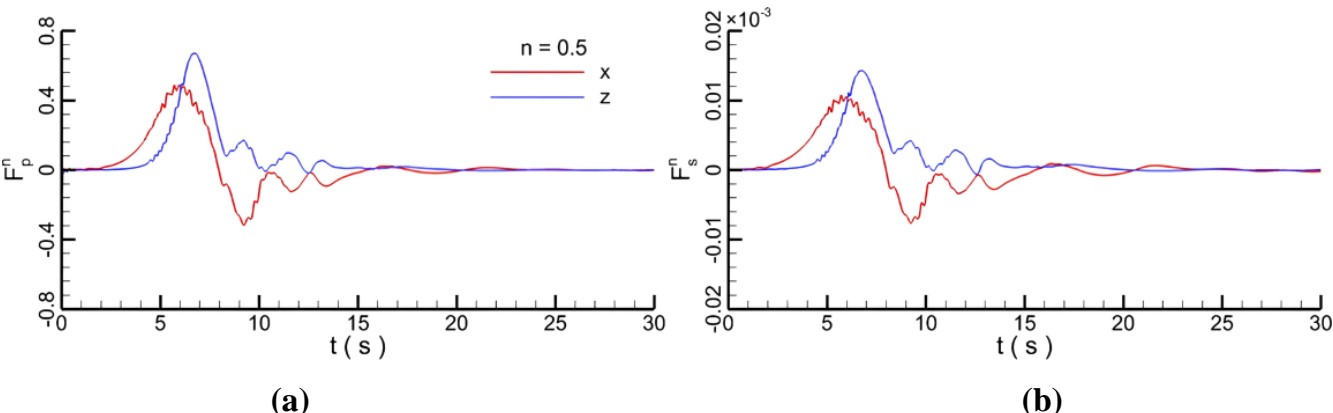

**(a)**                                                                 **(b)**

**Figure 11.** Decomposed pressure gradient force (**a**) and shear stress (**b**) force on the pipeline.

4.1.2. Thickness of Wrapper

Seven wrapper thicknesses are considered: $T$ = 0.2, 0.25, 0.3, 0.35, 0.4, 0.45, and 0.5 m. The porosity coefficient is taken to be 0.6. At the moment that the wave goes through the pipe, the transient evolution of the vorticity contours around the pipeline with a wrapper thickness of 0.25 m is depicted in Figure 12. A couple of vortices emerge on the upper and lower vertices of the pipeline as the wave approaches the pipeline. As the wave propagates, many vortices flow along the wrapper and then shed off. Compared with the top vortices, the bottom vortices are shed off faster for two reasons. Firstly, as the friction at the seabed is small, the bottom flow velocity is higher than that on the top. Secondly, when the wave peak departs from the pipeline, a strong disturbance by the water body occurs behind the pipeline, followed by the irregular swing and fall off of the vortices. After the wave travels far away, the water flow near the pipeline becomes weak, and the vortices are scattered around the pipeline.

Figure 13 shows a comparison of flow field stream traces and the velocity contours. When the fluid penetrates the wrapper, the streamline starts to diverge, which indicates that the free flow is hindered. Therefore, the flow becomes slower and the flow direction becomes non-uniform. For the fluid flows out of the wrapper, the stream traces are quite complex and chaotic. The reason is that the seeping fluid mixes with the bypass flows and causes strong interference in the water body behind the pipeline. The streamlines passing through the wrapper indicates frequent water exchange at the wrapper surface. Along with the small-attached vortices on the wrapper surface, more fluid passes over the wrapper and causes a large vortex behind the wrapper.

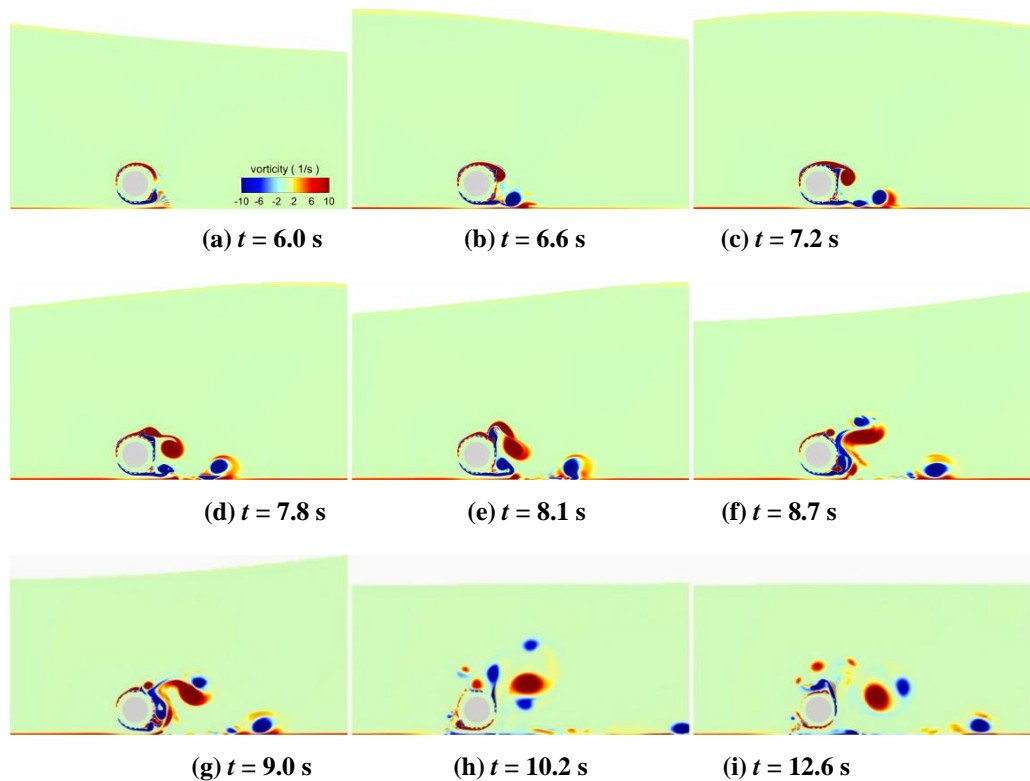

**Figure 12.** Temporal evolutions of vorticity contours around pipeline with wrapper thickness of 0.25 m (**a**) 6.0 s (**b**) 6.6 s (**c**) 7.2 s (**d**) 7.8 s (**e**) 8.1 s (**f**) 8.7 s (**g**) 9.0 s (**h**) 10.2 s (**i**) 12.6 s.

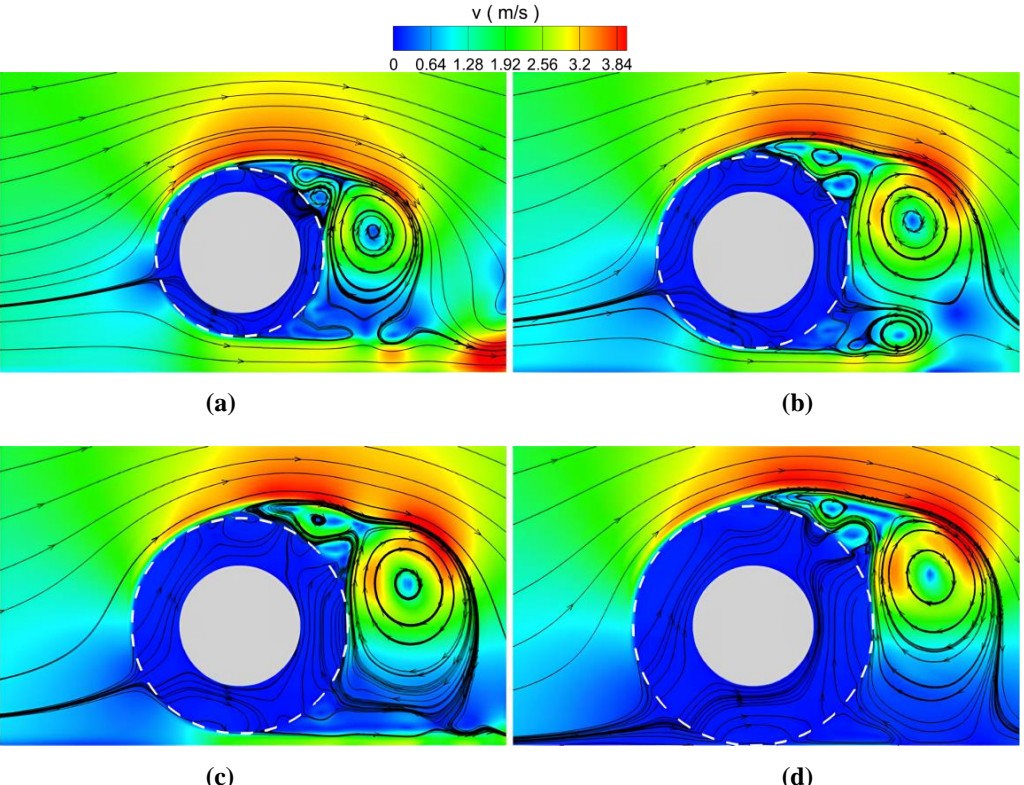

**Figure 13.** Comparisons of flow field streamtraces and velocity contours under different wrapper thicknesses; (**a**) T = 0.2 m; (**b**) T = 0.3 m; (**c**) T = 0.4 m; (**d**) T = 0.5 m.

The highest free surface elevations and velocities at the front and at the rear of the pipeline with different wrapper thicknesses are depicted in Figure 14. As the wrapper thickness increases, the highest elevation at the front of the pipeline seems to be quite stable, although the peak velocity increases by around 6%. At the moment that the wave bypasses the pipeline, the maximum elevation reduces with an increase in the wrapper thickness. This is because the pipeline blocks the wave propagation. However, due to the strong mixing effect of the seepage and bypass water, the maximum velocity rises to be higher than that in front of the pipe. The maximum forces on the wrapper and the pipeline for different wrapper thicknesses are shown in Figure 15. With an increase in the wrapper thickness from 0.2 to 0.5 m, the normalized forces on the wrapper are doubled as a larger interaction area is involved. In contrast, the vertical forces on the pipeline decrease by 12.5%. Therefore, the larger the thickness of the wrapper the safer the pipeline.

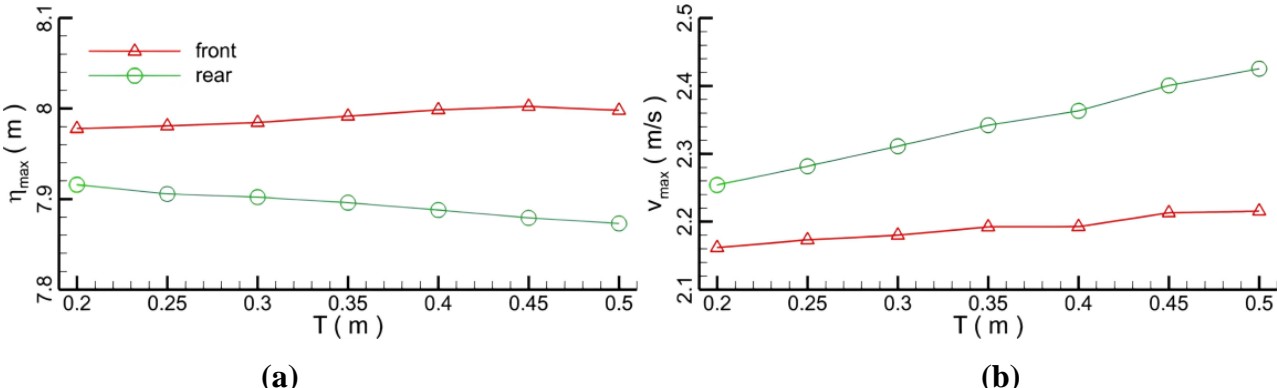

**Figure 14.** Comparisons of the maximum elevations and velocities in front and rear of the pipelines with different wrapper thicknesses; (**a**) free surface elevation (note: original water depth is 6 m); (**b**) velocity.

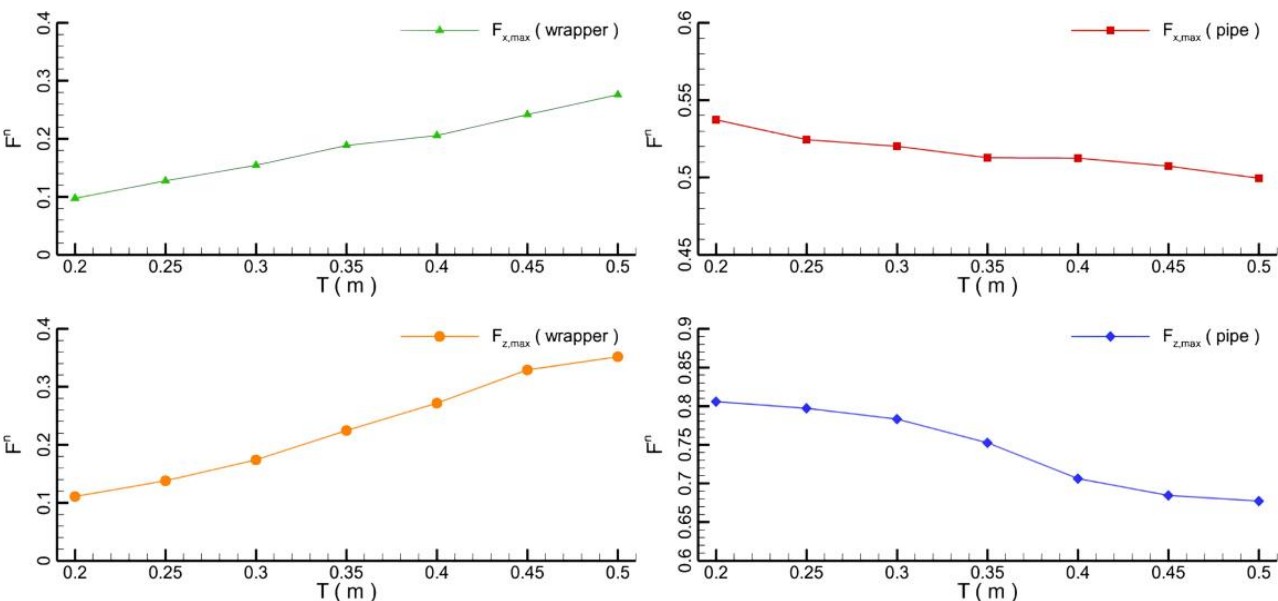

**Figure 15.** Hydrodynamic forces on the pipeline and wrapper.

### 4.2. Effect of Pipeline Structure

The in-situ pipelines may be under various suspended conditions since the seabed topography is often uneven. Some pipelines are also laid in tandem for the sake of the transportation efficiency. In order to examine the effects of porous wrappers on pipelines under different conditions, a study was carried out considering two scenarios, namely,

suspended pipelines and pipelines in tandem. In the numerical models, the porosity coefficient (*n*) remained at 0.6, the thickness (*T*) of the wrapper was kept at 0.5 m, and the wave height (*H*) was assumed to be 2.0 m.

### 4.2.1. Suspended Pipelines

Six gaps (*G*) between the wrapper and the seabed (0.0, 0.2, 0.4, 0.6, 0.8, and 1.0 m) were considered [35–37]. The representative flow field at three points in time (6.3, 7.2, and 10.2 s) are shown in Figure 16. At the arrival of the wave at the pipeline (at 6.3 s), the flow is accelerated and the velocities over and beneath the pipe reach the maximum values due to the bypass effect of the fluid. At the moment that the wave peak is above the pipe (at 7.2 s), all the velocities around the pipe reach their highest values. After the wave passes over the pipe (at 10.2 s), the velocity decreases and several vortices are formed behind the pipeline. With a tiny wrapper-seabed gap, the velocity within the gap is very high while the flux is relatively small. An increase in the gap will result in an increase in the flux and a decrease in the velocity. A symmetric velocity distribution similar to a fisheye is observed behind the pipeline, which becomes more obvious when the gap increases (Figure 16c).

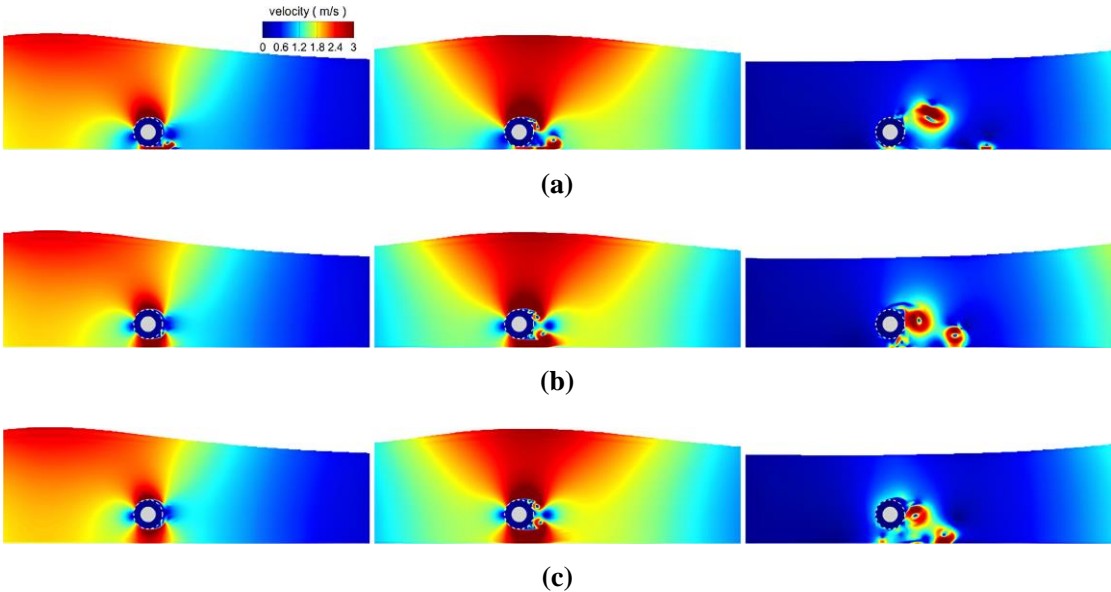

**Figure 16.** The velocity contours of the flow fields under different gaps; (**a**) G = 0.2 m; (**b**) G = 0.6 m; (**c**) G = 1.0 m. Left to right: 6.3 s, 7.2 s, and 10.2 s. Left to right: arrival, stay, departure.

With the bypass of the wave, the vortices generated around the pipeline become larger. The vorticity contours and the streamlines of the flow field are shown in Figure 17. As the solitary wave approaches, a pair of whirlpools shed off from the wrapper with a gap of 0.2 m. With an increase in the gap, the two whirlpools gradually disappear and are replaced with two smaller vortices. Due to the internal pores within the wrapper, the streamlines in the wrapper are dispersed, and it is hard for a vortex to be generated. With an increase in the gap, two anti-symmetric vortices shed off from the wrapper. Besides, some tiny vortices remain adhered to the wrapper due to the interaction by the seepage and the external flow. When the gap is very small, a few small vortices are generated between the wrapper and the seabed. In contrast to the interface of vortex from the flow around a solid cylinder, the vortex interface at the wrapper is not fully smooth. Because of the strong interactions of fluid over the wrapper surface, several small vortices mingle with the large shedding vortices. The flow direction also varies greatly according to the streamline mobilisation.

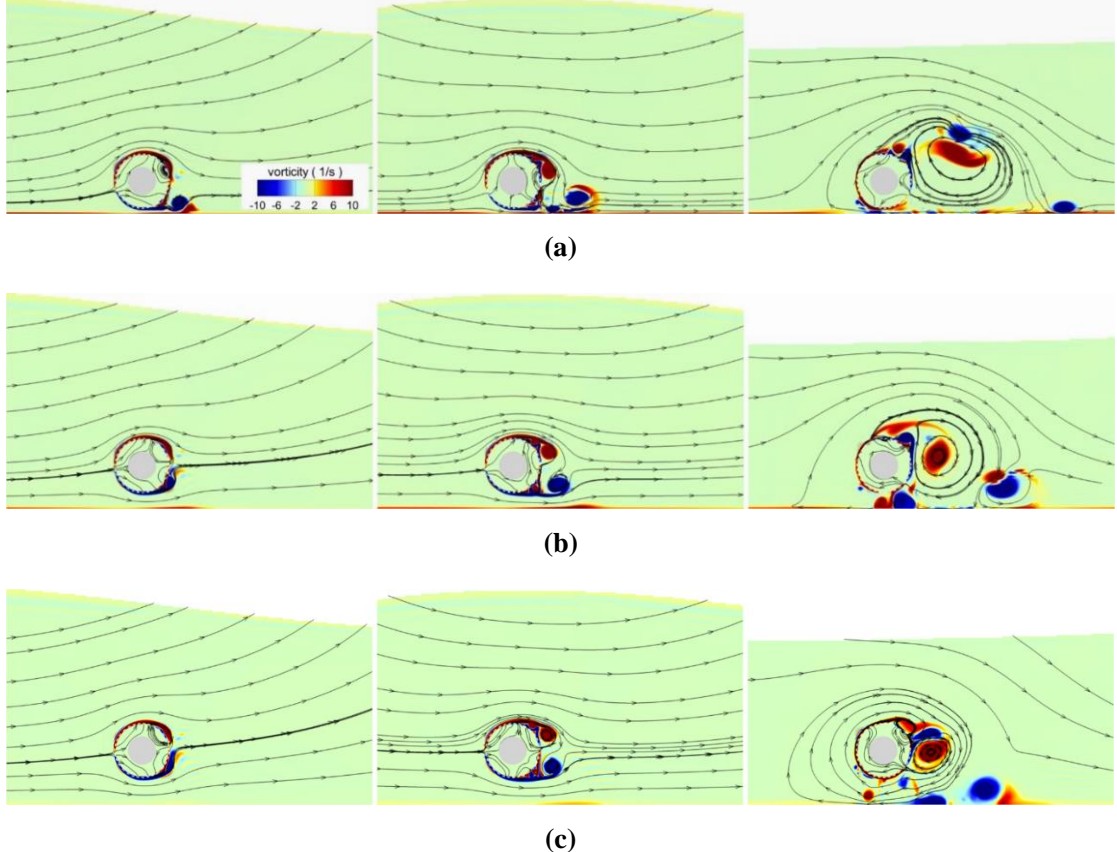

**Figure 17.** The vorticity contours of the flow fields under different gaps; (**a**) G = 0.2 m; (**b**) G = 0.6 m; (**c**) G = 1.0 m. Left to right: 6.3 s, 7.2 s and 10.2 s.

The gap is normalized by the pipeline diameter as $\beta = G/D$. With a small gap ($\beta < 0.2$), the horizontal forces on both the wrapper and the pipeline are slightly smaller than those on the wrapper and pipeline without a gap (Figure 18). With a further rise of the gap width, the horizontal forces are accordingly enlarged due to higher velocity around the pipeline as shown in Figure 16.

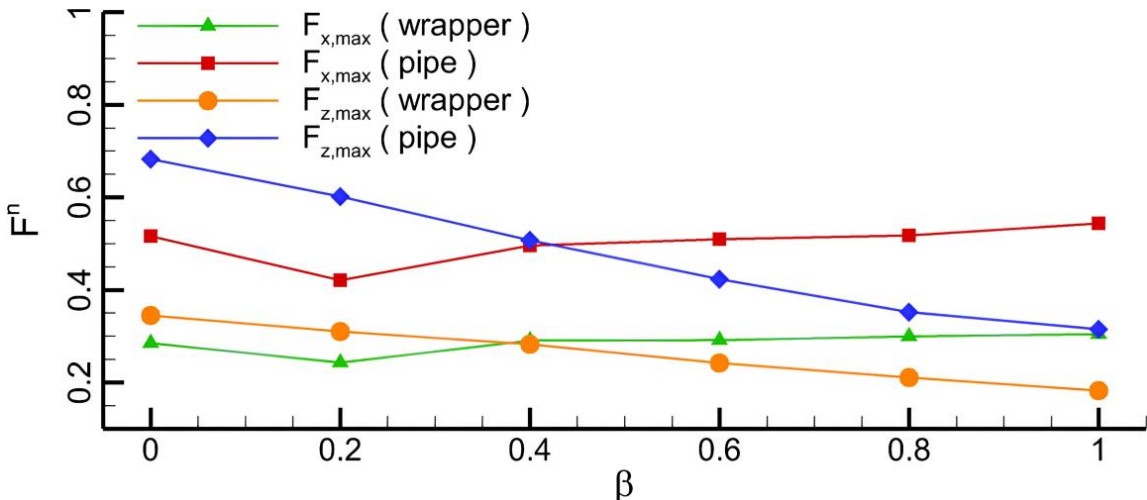

**Figure 18.** Comparisons of the maximum horizontal and vertical hydrodynamic forces on the pipeline and wrapper under different gaps.

In contrast, an increase in the gap width may inversely cause the reduction of vertical forces on both the wrapper and the pipeline. The vertical forces can be considered to consist of two parts. One is caused by the weight of the water body at the bypass of the wave from the pipeline, while the other can be caused by the velocity difference between the flow above and below the pipeline after the flow passes over. In summary, as the gap increases, the flow velocity within the gap initially increases when $\beta < 0.2$ and then decreases when $\beta > 0.2$. In contrast, the vertical forces caused by the wave's weight always decrease with an increase in the gap.

### 4.2.2. Pipelines in Tandem

The hydrodynamic forces on pipelines in tandem are investigated considering five different distances (*S*) between the two pipeline centres (2.5, 3.0, 3.5, 4.0, and 4.5 m). The velocity and vorticity fields at 6.3, 7.2, and 10.2 s around the tandem pipelines with distances of 2.5, 3.5, and 4.5 m are depicted (Figures 19 and 20). As the wave approaches the pipeline, the velocity within the pipeline gap is very small due to the blockage effect of the pipeline in front. As the distance increases, the velocity field within gap space is enhanced as more water flow is allowed. The velocity above the pipeline has its maximum value, and part of the high-speed fluid flows into the gap through the space underneath the pipeline. With a small distance, the vortices shedding off from the front pipeline impinge directly on the rear pipeline without any stretching. When the distance is increased, noticeable vortex shedding emerges in the middle space (Figure 20c). Similar vortex shedding behind the rear pipeline is observed for different distances. After the wave bypasses the pipeline, the increase in the distance between the pipelines will cause an increase in the velocity magnitudes in the space among the pipelines. As the distance increases, the flow becomes more chaotic due to the seepage from the wrapper and the limited flow space. In summary, influence of the distance between the pipelines over the whole kinematic field is not significant, although the local flow field around the pipelines is severely affected. When the wave bypasses the tandem pipelines, the largest forces on structures (i.e., the pipelines and wrappers) are shown in Figure 21, in which the distance ratio ($\theta$) is calculated as $\theta = S/D$.

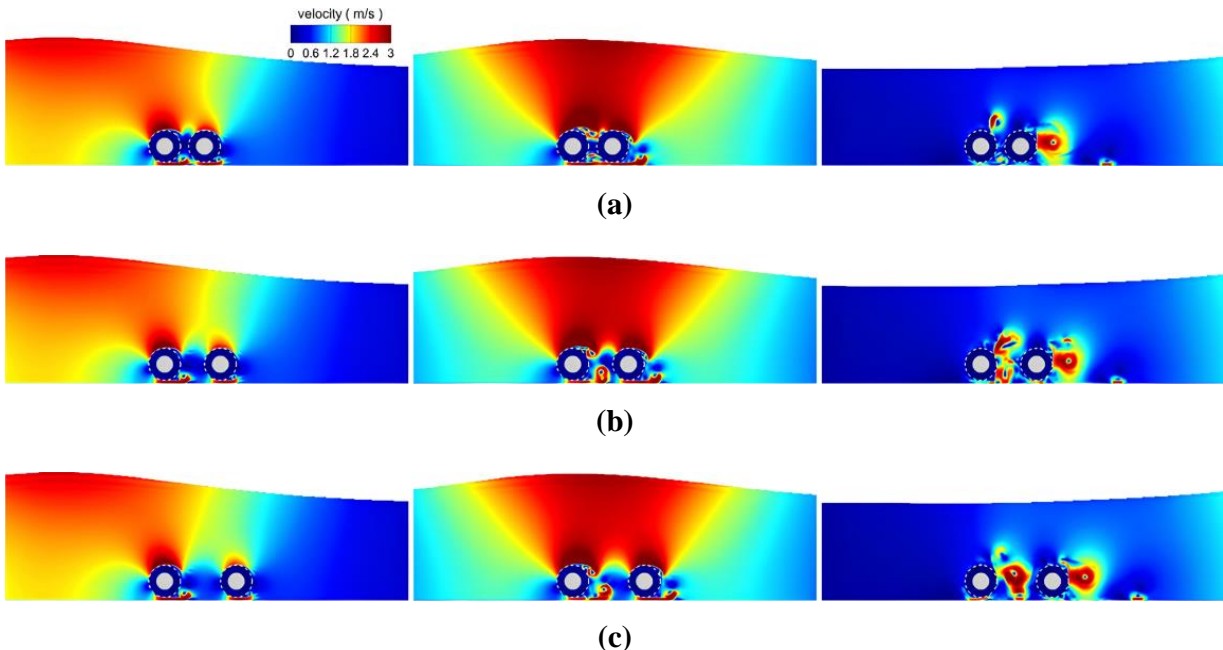

**Figure 19.** The velocity contours of the flow fields under different spacings; (**a**) S = 2.5 m; (**b**) S = 3.5 m; (**c**) S = 4.5 m. Left to right: 6.3 s, 7.2 s and 10.2 s.

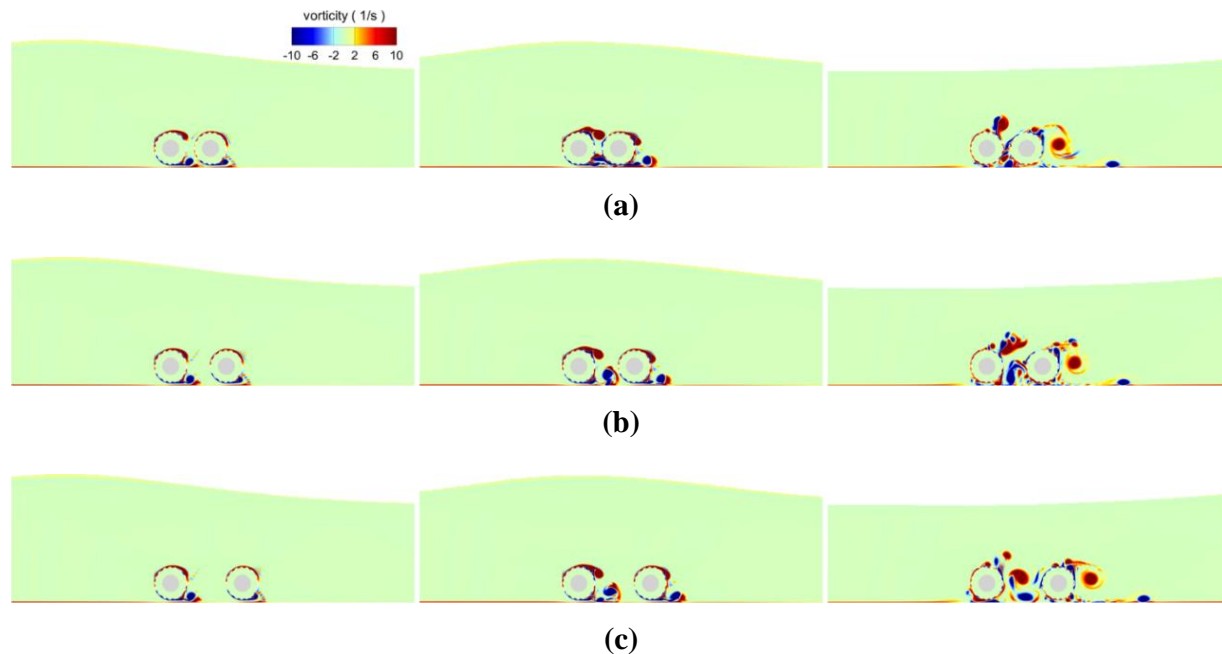

**Figure 20.** The vorticity contours of the flow fields under different porosities; (**a**) S = 2.5 m; (**b**) S = 3.5 m; (**c**) S = 4.5 m. Left to right: 6.3 s, 7.2 s and 10.2 s.

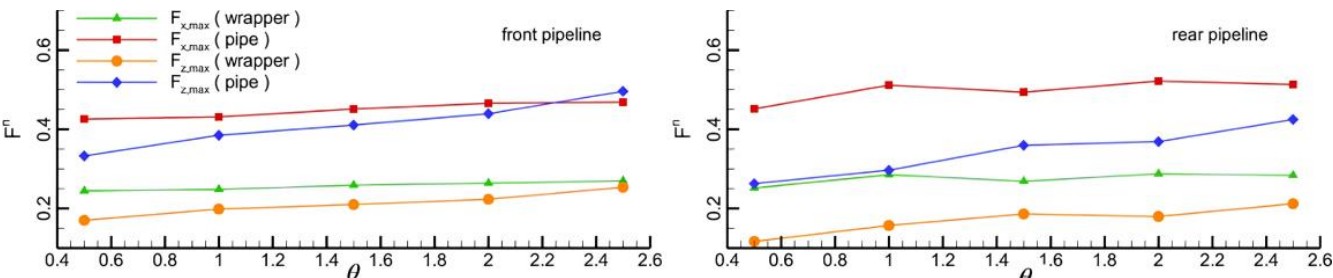

**Figure 21.** The maximum forces on the pipeline and wrapper under different distances.

As for the pipeline in front, as the distance ratio increases, the horizontal loads on the wrapper and pipeline increase slightly, while the vertical forces are almost doubled. As for the rear pipeline, as the distance reduces, the velocity in the gap becomes smaller and the forces on the pipelines and wrappers are also reduced, which is mainly attributed to the shield effect from the front pipeline. With an increase in the distance, the forces increase due to the increase in the turbulence energy in the gap.

Different ratios of the forces on the front and rear pipelines are depicted in Figure 22. The difference ratio is defined as $\Delta F^n = (f_{f,max} - f_{r,max})/f_{f,max}$, where $f_{f,max}$ and $f_{r,max}$ are the maximum forces on the pipeline or wrapper. It is found that the horizontal loads on the rear pipe and wrapper tend to be always higher than their counterparts on the front pipe. This means that a turbulent flow in the horizontal direction on rear pipe is more intense than that on the front pipe. For different distances, deviations for the forces on the pipelines and wrappers are also different. The deviation is found to be maximized at a distance of 1 m and this indicates that the pipeline is not well protected and needs to be avoided in engineering practice.

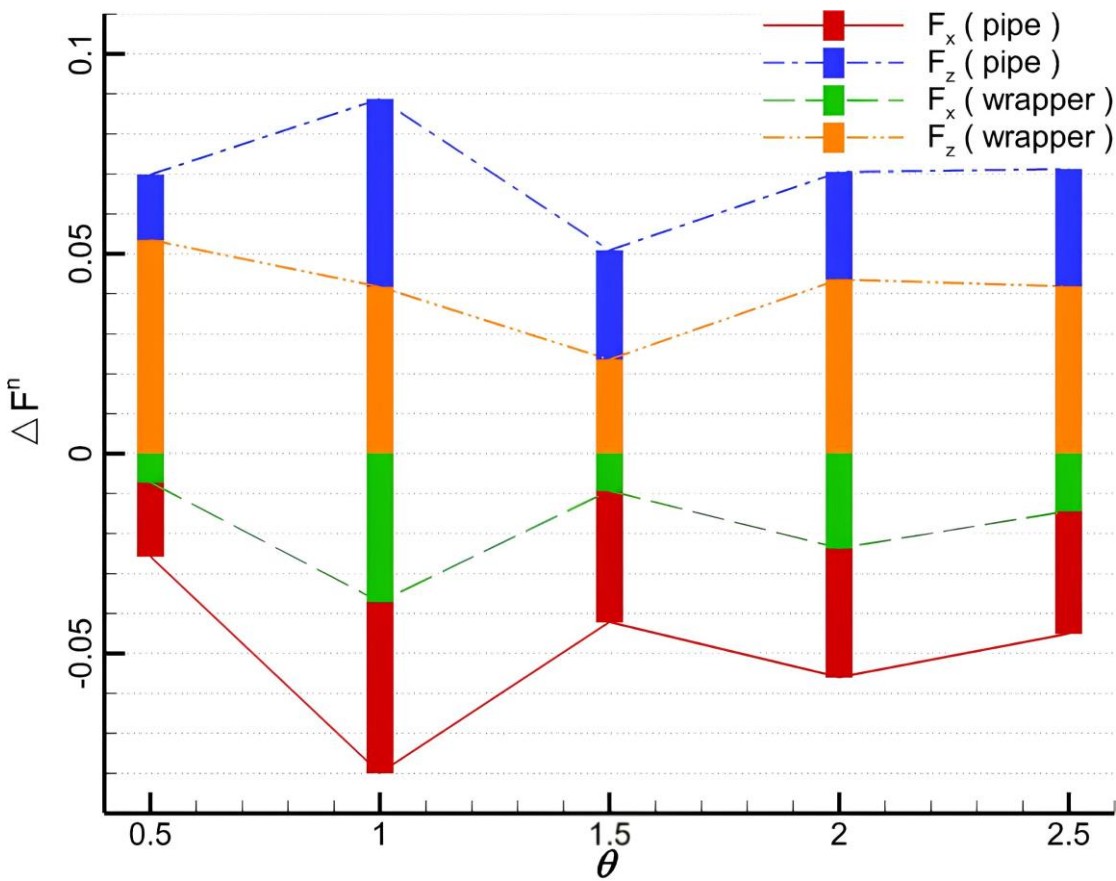

**Figure 22.** The deviation of the forces on the front and rear pipelines and wrappers under different distances.

### 4.3. Effect of Wave Height

Six groups of wave heights (*H*), i.e., 1.6, 1.8, 2.0, 2.2, 2.4, and 2.6 m, are selected to consider different marine environment. After bypassing the pipeline, the height of the wave decreases because of the blockage effect of the pipeline and the dissipation of the flow energy (Figure 23a). The deviation ratio of the wave heights before and after the wave passes over the pipeline is shown in Figure 23b and is defined as $\delta = (H_{f,max} - H_{r,max})/H_{f,max}$. The wave height attenuation becomes more significant as the wave height increases. This means that waves with larger heights are more easily affected by the pipelines.

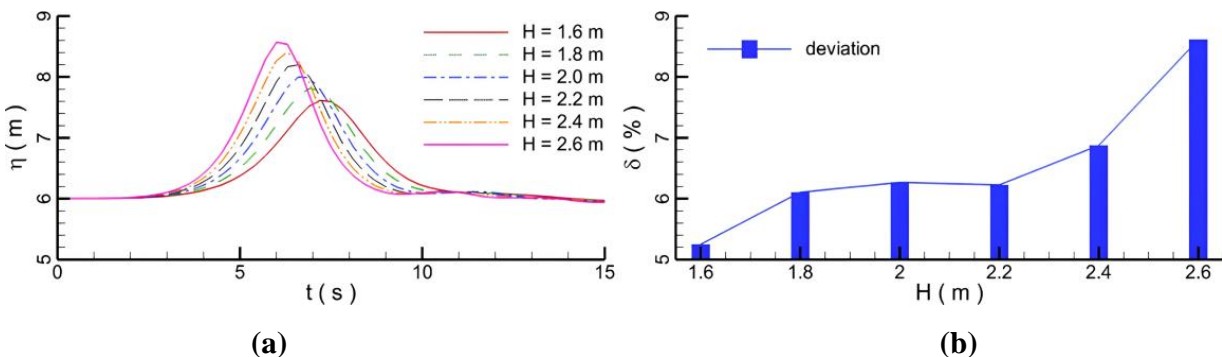

| (a) | (b) |

**Figure 23.** Waves with different wave heights; (**a**) temporal evolutions; (**b**) attenuation deviation.

At the bypass of the wave through the pipe, the loads are increased until they reach the maximum values at the moment that the wave peak appears above pipeline (Figure 24). The

forces gradually decrease as the wave propagates. Because of some reflux after the wave bypasses the pipeline, the flow is in the opposite direction to that of the wave propagation, resulting in a negative force. The vibration of the water body by the wave propagation induces oscillations of the forces on the pipeline and wrapper. When the wave height is larger, the force oscillation becomes fiercer and the maximum loads on the pipeline and the wrapper increase (Figure 25). The vertical forces on the pipeline are the largest compared with other forces under the same conditions. Besides, as the wave height increases, the increased amplitude of vertical forces on the pipeline is the most significant change since the weight of the water above the pipeline increases. Therefore, given that the wave height is very high, the protective function of wrapper on the pipeline tends to be weakened compared with that of the wrapper for a low wave height.

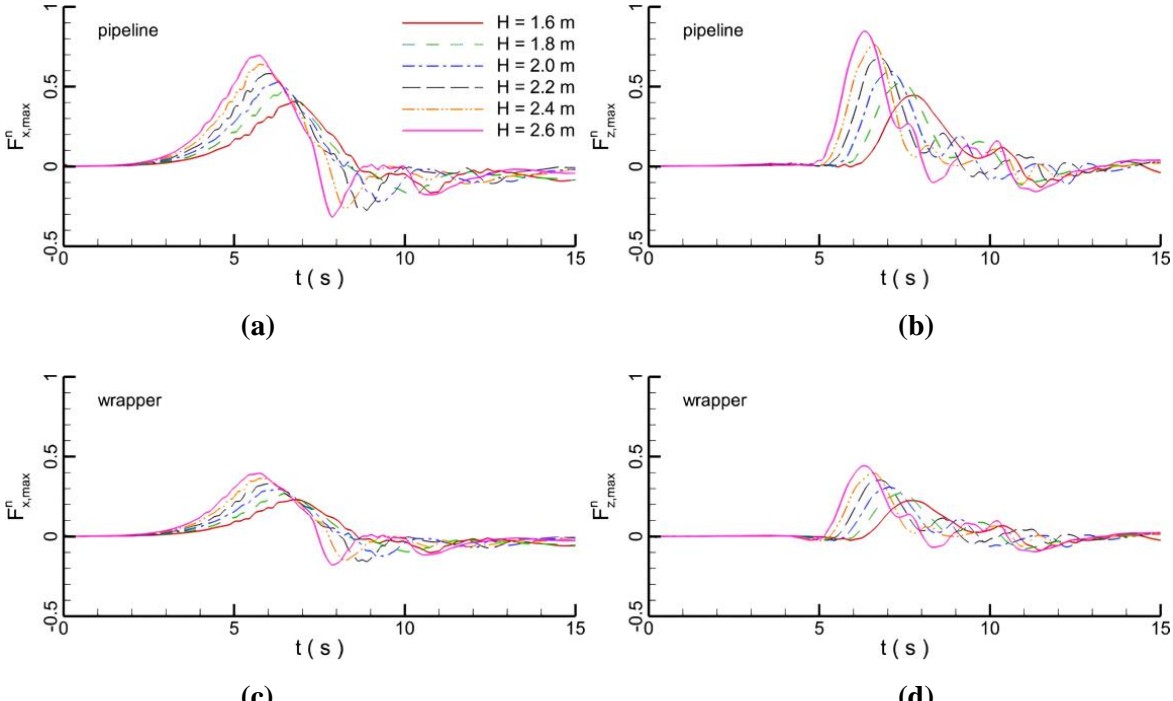

**Figure 24.** The temporal evolutions of forces on the pipeline and wrapper; (**a**) Horizontal maximum force on pipeline; (**b**) Vertical maximum force on pipeline; (**c**) Horizontal maximum force on wrapper; (**d**) Vertical maximum force on wrapper.

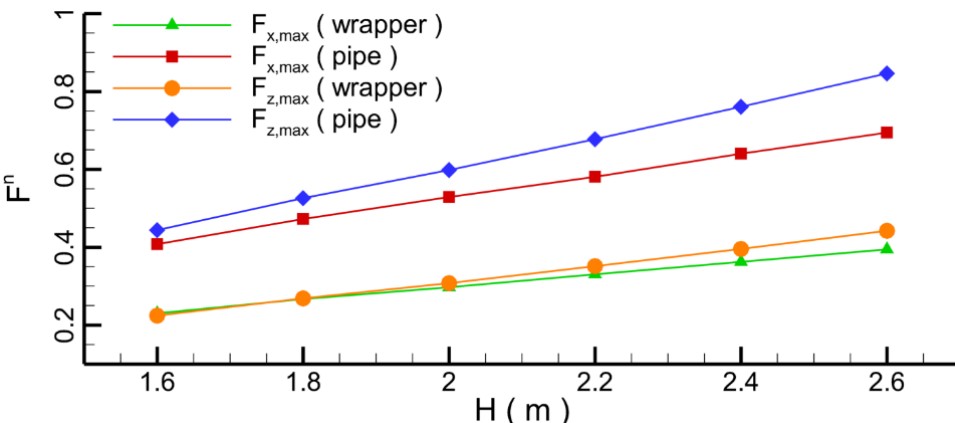

**Figure 25.** Variation of hydrodynamic forces on the pipeline and wrapper under different distances.

## 5. Conclusions

The effect of porous media on the dynamic performance of submarine pipelines under solitary waves was investigated. The porosity of the wrapper, the seabed topography, the structure of the pipeline, and the marine environment were considered. The study had a limitation of the model sizes due to the limited computational resource and the simplification of the solitary wave due to its mathematical complication, which will be tackled in future works. The following main conclusions have been made.

(1)    When a pipe is wrapped by a porous medium, the velocity in the wrapper is relatively small because the porous medium can consume the water energy and weaken the flow. With an increase in the porosity, the range of the low-speed flow at the bottom of the pipeline expands. This indicates that the porous wrapper can slow down the flow and affect a wider region of the surrounding water. After the bypass of the wave through the pipe, the number and volume of the vortices behind the porous wrapper are larger than those for a pipeline with a solid wrapper or without a wrapper. As the porosity coefficient increases, the impact forces on the pipe increase, while those on the wrapper decrease. This implies that the porous wrapper is capable of protecting the pipeline.

With an increase in the wrapper's thickness, the hydrodynamic forces on the wrapper tend to increase. In particular, the horizontal forces on the pipeline decrease with an increase in the thickness due to the protection of the wrapper, while the vertical forces are increased because of variations in the fluid's stagnation point.

(2)    For a wave bypassing a pipe with different heights, a symmetric speed change similar to a fisheye appears behind the pipeline, along with two antisymmetric vortices shedding off from the wrapper.

As the internal seepage interacts with the external fluid flow, several small vortices are still attached to the wrapper. The hydrodynamic vertical forces on both the wrapper and the pipeline decrease with the pipeline distance. With an increase in the suspension of the pipe, the velocity and TKE within the gap space increase and both the vortex intensity and the number of vortices increase. Therefore, the flow pattern appears to be chaotic. As for the front pipeline, an increase in the gap leads to a slight increase in the horizontal forces on both the wrapper and the pipeline, but a significant increase in the vertical forces. As for the rear pipeline, because of the shield function of the front pipeline, the velocity within the gap space and the forces on the pipes decrease with a decrease in the gap size.

(3)    When the waves with different heights pass over the pipeline, the height of the wave is reduced because of the blockage function from the pipeline and the dissipation characteristic of the flow energy. When the wave height is increased, the velocity around the pipeline increases, inducing an increase in the TKE. As the wave height increases, all the maximum forces on the pipeline and wrapper also increase. Note that an increase in the vertical forces on the pipeline is the most significant change because the weight of the water above the pipeline increases, which implies that the protection function of the wrapper is enhanced by the reduction in the wave height.

From the above investigation, the mechanism of load transfer from the pipeline to the external wrapper has been presented. This encourages industrial experts and academic scholars to arrange more investigations of the functions and cost-efficiency of porous wrappers, which could form a new branch of the pipeline design practice.

**Author Contributions:** Contributor Roles Taxonomy: E.Z.: Conceptualization, Methodology, Validation, Investigation and Writing—Original Draft; Y.D.: Data Curation, Formal analysis; Y.D.: Visualization, Project administration; Y.D., L.C., Y.W. and Y.L.: Writing—Review & Editing. All authors have read and agreed to the published version of the manuscript.

**Funding:** The paper was supported by the National Natural Science Foundations of China (Grants No. 52001286 and No. 42272328), GuangDong Basic and Applied Basic Research Foundation (Grant No. 2022A1515240002) and Comprehensive Survey of Natural Resources in Huizhou-Shanwei Coastal Zone (Grant No. DD20230415).

**Data Availability Statement:** The data that support the findings of this study are available from the corresponding author upon request.

**Conflicts of Interest:** The authors declare that they have no conflict of interest.

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
