# Peer review of "Dynamic Performance of Suspended Pipelines with Permeable Wrappers under Solitary Waves"

_jmse, doi:10.3390/jmse11101872_

Round 1
Reviewer 1 Report
3.1. Propagation Over a Porous Breakwater
…FLOW-3D are close to both the experimental measurements and the numerical predictions from another CFD software package ANSYS… More details are needed. Name the CFD codes (with references) and the version of ANSYS. More details are needed here to increase the quality of the manuscript.
3.2. Forces on pipeline
…with L as the per unit width… or the length. Give more details for the value of L.
Fig. 3: Why don't you have all the comparisons for x=0.04, 0.08 and 0.12 m? Give an explanation.
Fig. 4: Ηere, we observe a deviation in experiment and simulation comparisons between 2.5 and 3 s. Why is this? Give an explanation.
Fig. 9: a better analysis for Fig. 9 is needed.
General comments
More details for the theoretical background for the commercial package FLOW-3D are needed.
More details for the Neumann-type absorbing boundary condition are needed.
-
Reviewer 2 Report
- The subject addressed is within the scope of the journal.
- However, the manuscript, in its present form, contains several weaknesses. Appropriate revisions to the following points should be undertaken in order to justify recommendation for publication.
- The abstract could become much better if it properly introduces the study from a research standpoint. Also, the main findings could be stated more pointedly in the abstract.
- The authors should consider the effect of wave reflection on the results.
- Some pictures should be presented to show the experimental procedure.
- The quality of the figures is weak. The original source of the figures should be used in the manuscript.
- It is suggested to discuss about scale effects in the present work (authors can use articles entitled “Safari Ghaleh et al. Numerical Modeling of Failure Mechanisms in Articulated Concrete Block Mattress as a Sustainable Coastal Protection Structure”).
- More explanations should be added for the meshing conditions (along with a figure).
- All symbols and parameters should be defined, please check.
- Some key parameters are not mentioned. The rationale on the choice of the particular set of parameters should be explained with more details. Have the authors experimented with other sets of values? What are the sensitivities of these parameters on the results?
- Draw a flowchart from your workflow that briefly shows the process of the methodology.
- It is suggested to add articles entitled “Widyastuti et al. Dam-Break Energy of Porous Structure for Scour Countermeasure at Bridge Abutment” and “Cuong & Tuan. Wave Hydrodynamics across Steep Platform Reefs: A Laboratory Study” to the literature review.
- Conclusion:
•The conclusion section is currently a repeat or rehash of the preceding sections, and needs to be re-written to improve it, keeping in mind the following suggestions.
•Update the conclusion to include the newly formulated theoretical contributions
•Mention the limitations of the study and prospects for future research.
•Summarize the key results in a compact form and re-emphasize their significance.
•This conclusion could be worded in a manner as to emphatically motivate the academic community to get down to actionable, practical engaged scholarship.
- Using the article entitled "Yamini et al. Hydraulic Performance of Seawater Intake System Using CFD Modeling", discuss regarding the vorticities and vibration in pipes.
- Page 16: the following paragraph is unclear, so please reorganize that:
“At the moment that the wave has gone through the pipeline, a growth of the distance will cause an increase in the velocity filed in the middle space, thus strengthening the flow intensity. As the distance increases, the flow pattern becomes more chaotic because of the seepage from the wrapper and the limited flow space. In summary, influence of the pipeline distance over the whole kinematic field is not significant, although the local flow field around the pipelines is affected. At the moment that the wave bypasses the tandem pipelines, the largest forces upon structures (i.e. the pipelines and wrappers) are shown in Fig. 21, in which the distance ratio (q) is calculated as q = S/D.”.
Round 2
Reviewer 2 Report
The article has been revised very well, so I would suggest to accept in its present form.
Author Response
Thanks for the reviewer's approval.